# The Role of Propionate-Induced Rearrangement of Membrane Proteins in the Formation of the Virulent Phenotype of Crohn’s Disease-Associated Adherent-Invasive *Escherichia coli*

**DOI:** 10.3390/ijms251810118

**Published:** 2024-09-20

**Authors:** Olga V. Pobeguts, Maria A. Galyamina, Elena V. Mikhalchik, Sergey I. Kovalchuk, Igor P. Smirnov, Alena V. Lee, Lyubov Yu. Filatova, Kirill V. Sikamov, Oleg M. Panasenko, Alexey Yu. Gorbachev

**Affiliations:** 1Lopukhin Federal Research and Clinical Center of Physical-Chemical Medicine of Federal Medical Biological Agency, Malaya Pirogovskaya 1a, 119435 Moscow, Russialemik2007@yandex.ru (E.V.M.); lee_1604@mail.ru (A.V.L.); sikamov2000@gmail.com (K.V.S.); o-panas@mail.ru (O.M.P.); augorbachev@gmail.com (A.Y.G.); 2Shemyakin-Ovchinnikov Institute of Bioorganic Chemistry, Ulitsa Mikluho-Maklaya, 16/10, 117997 Moscow, Russia; 3Department of Chemistry, Lomonosov Moscow State University, Leninskiye Gory 1–3, 119991 Moscow, Russia; luboff.filatova@gmail.com

**Keywords:** Crohn’s disease, Crohn’s disease-associated adherent-invasive *Escherichia coli* (AIEC), propionic acid, neutrophiles, membrane proteins, 2D electrophoresis, LC-MS analysis

## Abstract

Adhesive-invasive *E. coli* has been suggested to be associated with the development of Crohn’s disease (CD). It is assumed that they can provoke the onset of the inflammatory process as a result of the invasion of intestinal epithelial cells and then, due to survival inside macrophages and dendritic cells, stimulate chronic inflammation. In previous reports, we have shown that passage of the CD isolate ZvL2 on minimal medium M9 supplemented with sodium propionate (PA) as a carbon source stimulates and inhibits the adherent-invasive properties and the ability to survive in macrophages. This effect was reversible and not observed for the laboratory strain K12 MG1655. We were able to compare the isogenic strain AIEC in two phenotypes—virulent (ZvL2-PA) and non-virulent (ZvL2-GLU). Unlike ZvL2-GLU, ZvL2-PA activates the production of ROS and cytokines when interacting with neutrophils. The laboratory strain does not cause a similar effect. To activate neutrophils, bacterial opsonization is necessary. Differences in neutrophil NADH oxidase activation and ζ-potential for ZvL2-GLU and ZvL2-PA are associated with changes in membrane protein abundance, as demonstrated by differential 2D electrophoresis and LC-MS. The increase in ROS and cytokine production during the interaction of ZvL2-PA with neutrophils is associated with a rearrangement of the abundance of membrane proteins, which leads to the activation of Rcs and PhoP/Q signaling pathways and changes in the composition and/or modification of LPS. Certain isoforms of OmpA may play a role in the formation of the virulent phenotype of ZvL2-PA and participate in the activation of NADPH oxidase in neutrophils.

## 1. Introduction

Adhesive-invasive *Escherichia coli* (AIEC) was first isolated from the ileal mucosa of a patient with Crohn’s disease (CD), which is a severe chronic immune-mediated inflammatory disease of the gastrointestinal tract. It turned out that they are able to successfully penetrate the mucin layer, overcome the epithelial barrier, and also survive and multiply inside macrophages [1,2] and dendritic cells [3,4]. Bacteria with such properties were assigned to a special group of pathobiont adhesive-invasive *E. coli* [5,6,7]. AIEC activity is accompanied by the release of pro-inflammatory cytokines [8], i.e., surviving and multiplying inside macrophages; they enhance the inflammatory process. The role of AIEC in the onset or chronicity of CD is not well defined. However, it has been proposed that these bacteria could trigger the onset of the inflammatory process as a result of the invasion of intestinal epithelial cells, and then, due to their survival within macrophages, they could stimulate chronic inflammation and granuloma development [2].

As soon as A. Darfeuille–Michaud et al. identified the AIEC pathotype in 2004 [2], the search for unique genes that could explain this phenotype began. In 2010, the first AIEC genomes were sequenced, and since then, many comparative genomic studies have been carried out in an attempt to elucidate the characteristics of the AIEC genome and identify a genetic biomarker [9,10]. However, no gene or gene sequence has yet been identified that is unique to the AIEC pathotype. Only those genes that were more abundant in AIEC strains compared to non-invasive *E. coli* strains were identified. Among them are genes associated with capsule formation, propanediol utilization, pili components, and iron retention [11,12,13,14,15,16]. An analysis of single nucleotide polymorphisms (SNPs) in AIEC genomes was also performed. However, specific SNPs for the AIEC pathotype have not been identified. It is suggested that changes in the sequence of the three genes *fimH*, *chiA*, and *ompA* may be associated with the virulence properties of AIEC [17,18,19]. The mechanisms that are responsible for the ability to attach and penetrate epithelial cells, as well as to survive and replicate in macrophages, are still poorly understood. AIEC do not have genetic invasive determinants characteristic of enteroinvasive, enteropathogenic, or enterotoxigenic *E. coli* [1,20]. Only a few virulence factors have been identified. Among them are the flagellum and type I pili, chitinase ChiA, outer membrane porins OmpA and OmpW (they are involved in adhesion invasion) [21,22,23], as well as the *ibeA* gene, which plays a role in the interaction between epithelial cells and macrophages and also promotes colonization of the mouse intestine [24].

Pathogens use a variety of mechanisms, including the induction of inflammation, the direct or indirect destruction of commensal species, and the use of alternative carbon sources to survive [25,26]. AIEC are shown to be able to utilize ethanolamine and propanediol, which are formed during the catabolism of phospholipids, fucose or rhamnose, propionate, and other metabolites [23,27,28]. Metabolic plasticity is thought to allow AIEC to act as an opportunistic pathogen in conditions of intestinal inflammation. We have previously shown that passage of AIEC from a CD patient (CD isolate) on M9 minimal medium supplemented with sodium propionate (PA) as a carbon source leads to a strong increase, and passage on M9 medium supplemented with glucose, on the contrary, leads to a significant decrease in adhesive-invasive properties and ability to survive in macrophages [23]. We were able to compare the isogenic CD isolate in two states: virulent with high adhesive-invasive activity and ability to survive in macrophages, and non-virulent when these properties are lost. In contrast to the CD isolate, passage of the laboratory strain K12 MG1655 on the M9 medium supplemented with PA did not cause a similar effect.

Neutrophils are detected in the inflamed colon in Crohn’s disease, and the percentage of neutrophils is correlated with the endoscopic score [29]. Both increases in neutrophil carcinoembryonic antigen-related cell adhesion molecule 6 CD66b (in mucosa) and CD64 (in blood) were registered with disease aggravation. CD66b (CEACAM8) has been reported to participate in neutrophil adhesion [30]. AIEC are known to bind to CEACAM8 via the FimH protein, which is a component of type I pili [17]. CD64 is the high affinity Fc receptor for IgG, and its expression on circulating neutrophils might increase during any infectious process [31]. Recruitment of neutrophils is necessary for killing invading pathogens via release of granule contents, phagocytosis with respiratory burst, and generation of extracellular traps. In this work, we performed a comparative analysis of the production of reactive oxygen species and cytokines by neutrophils during their interaction with virulent and non-virulent phenotypes of the CD isolate ZvL2 (ZvL2-PA and ZvL2-GLU, respectively). Membrane proteins are the first to interact with innate immune cells. Therefore, we performed a comparative proteomic analysis of membrane fractions isolated from ZvL2-PA and ZvL2-GLU using 2D-DIGE and liquid chromatography–mass spectrometry (LC-MS). The laboratory strain K12 MG1655 was used as a control.

## 2. Results and Discussion

### 2.1. Kinetics of ROS Production by Isolated Human Blood Neutrophils upon Activation by a CD Isolate ZvL2 and a Laboratory Strain MG1655 without and in the Presence of Autologous Serum

Preliminary cellular cultures of CD isolate ZvL2 and laboratory strain K12 MG1655 were obtained and subjected to five passages in M9 medium supplemented with PA (ZvL2-PA and MG1655-PA) or glucose (ZvL2-GLU and MG1655-GLU). The activation of isolated neutrophils was assessed using lucigenin-dependent chemiluminescence (Luc-CL), which is sensitive to superoxide anions. Figure 1 shows typical time curves of the Luc-CL response of neutrophils stimulated with cells of virulent (ZvL2-PA) and non-virulent (ZvL2-GLU) phenotypes. Experiments were conducted both in the absence and presence of 2% autologous serum. Without serum addition, both ZvL2-PA and ZvL2-GLU exhibited low capacity to induce neutrophil activation. The amplitude and total luminescence values of Luc-CL in response to the addition of CD isolate passaged in glucose were 8.9 ± 1.9 V and 1538 ± 130 V, respectively, approximately twice as high compared to those of the isolate passaged in PA (5.0 ± 1.1 V and 598 ± 72 V, respectively). When autologous serum was added to isolated neutrophils, the amplitude and total luminescence values of the Luc-CL response significantly increased. For ZvL2-PA, they reached 64.6 ± 6.8 V and 7880 ± 133 V, respectively, whereas for ZvL2-GLU, these values were significantly lower, averaging 19.1 ± 8.7 V and 4689 ± 614 V. As a control, similar experiments were conducted for the laboratory strain K12 MG1655 (Figure 1B). In this case, we also observed weak activation of isolated neutrophils without serum addition (amplitude and total luminescence values in response to MG1655 cells passaged in glucose (MG1655-GLU) and PA (MG1655-PA) were 8.4 ± 1.3 V; 343 ± 37 V and 8.4 ± 0.67 V; 187 ± 14 V, respectively). In the presence of 2% autologous serum, the amplitude and total luminescence values increased and reached 33.8 ± 8.6 V and 5280 ± 450 V for MG1655-GLU activation, and 30.6 ± 4.0 V and 5689 ± 1050 V for MG1655-PA activation. The results indicate that interactions between bacteria and neutrophils occur even in the absence of serum; however, opsonization of bacteria enhanced the activation of NADH-oxidase. The mean values of amplitude and total luminescence observed for MG1655 did not depend on the carbon source used for culture growth. In the case of the CD isolate, these values were three times higher for ZvL2-PA compared to ZvL2-GLU. These data are consistent with our previous results, which showed that the virulence properties of ZvL2-PA, namely adhesive-invasive potential and survival in macrophages, were significantly increased, whereas they were low in ZvL2-GLU [21]. The outer membrane of Gram-negative bacteria, such as *E. coli*, plays a central role in controlling bacterial interactions with the external environment. One of the reasons for the significant differences in neutrophil activation by ZvL2-PA and ZvL2-GLU cells, as well as between ZvL2-PA and MG1655-PA cells, could be the structure of lipopolysaccharides (LPS) and changes in the repertoire and abundance of outer membrane proteins, particularly receptor-binding antibodies or complement components.

### 2.2. Production of Cytokines TNF-α, IL-6, and IL-1β by Isolated Human Blood Neutrophils after Incubation with CD Isolate ZvL2 and Laboratory Strain MG1655 In Vitro and ζ-Potential

To assess cytokine production, human blood neutrophils were incubated with ZvL2-PA and ZvL2-GLU cells, as well as with MG1655-GLU and MG1655-PA cells (Figure 1C). Cytokine production in the extracellular medium was determined using ELISA. Interaction of neutrophils with ZvL2-PA cells resulted in a 5-fold increase in IL-1β production and a 2-fold increase in TNF-α production compared to ZvL2-GLU. In the case of MG1655-PA, IL-1β production increased 2-fold, and TNF-α production increased 2.5-fold compared to MG1655-GLU. IL-6 production was approximately the same for both the CD isolate and the laboratory strain. The difference in the amount of cytokines produced by neutrophils when interacting with ZvL2 and MG1655 cells grown in M9 medium with the addition of PA or glucose may be related to the restructuring of the outer cell wall of bacteria, which occurs when switching from one carbon source to another. We also compared the ζ-potential values measured in cells of both strains grown on PA and glucose. These values differed significantly: bacteria grown on PA had more negative values than bacteria grown on glucose (Figure 1D).

### 2.3. Comparative Proteomic Analysis of Membrane Fractions Isolated from the Cells of the CD Isolate ZvL2 and Laboratory Strain MG1655

Membrane-enriched fractions were isolated from the CD isolate and laboratory strain cells after five passages in minimal M9 medium supplemented with glucose or PA. A comparative proteomic analysis of these membrane fractions was performed using two independent methods: two-dimensional electrophoresis (2D-DIGE) with differential staining and HPLC mass spectrometry.

#### 2.3.1. 2D-DIGE Shown that PA Causes a Significant Rearrangement in the Abundance of Membrane Proteins Compared to Glucose for Both the CD Isolate ZvL2 and the Laboratory Strain MG1655

The 2D maps are shown in Figure 2 and Figure 3, where arrows indicate differentially expressed proteins. Green spots on the 2D map correspond to membrane proteins, the level of which increases by 2 or more times, and red spots—decrease by 2 or more times when comparing isolates grown on PA relative to glucose. The cutoff score for protein identification in the mascot engine was 44 (*p* < 0.05). A protein was identified as significantly changed if the fold change was greater than 2. Quantitative analysis of differentially expressed proteins obtained as a result of 2D-DIGE was shown in Table 1 and Table 2. The most represented functional groups among the changing proteins were porins, transporters, proteins of biosynthesis and binding of the siderophore, proteins of cell division, LPS biosynthesis, amino acid biosynthesis, and peptide recycling.

Passage of both the CD isolate ZvL2 and the laboratory strain MG1655 in minimal medium supplemented with PA resulted in significant changes in the abundance of membrane proteins compared to growth on glucose. These changes were mostly similar for both strains. The abundance of porin OmpW and one isoform of OmpF significantly increased, while the abundance of porin OmpX decreased in both cases. It is known that the expression of the OmpX gene is associated with the formation of a flagellar apparatus [32], and it decreases during starvation [33], when *E. coli* loses its flagella as an energy-intensive complex. It is likely that under starvation conditions when growing on PA, both isolates may shed their flagella. For both isolates, the levels of EntB and EntE proteins involved in the biosynthesis of enterobactin, a siderophore that binds trivalent iron and maintains its homeostasis in the cell, increased. PA induced an increase in one isoform and a decrease in another isoform of the Fe-binding protein FepA, which binds ferrienterobactin and allows *E. coli* to extract iron from the environment. PA also caused an increase in the number of proteins involved in LPS synthesis. Common to both strains was the increase in mannose-1-phosphate guanylyltransferase ManC and UTP--glucose-1-phosphate uridylyltransferase GalU. PA also induced an increase in the abundance of transport proteins—outer membrane protein FadL (transport of long-chain fatty acids), LamB (transport of maltose and maltooligosaccharides), glutamine ABC transporter ATP-binding protein GlnQ, and outer membrane protein TolC (forms an outer membrane channel necessary for the functioning of several efflux systems). A common PA-induced protein in ZvL2-PA and MG1655-PA was rod shape-determining protein MreB, an actin-like protein necessary for maintaining rod shape and ensuring uniform peptidoglycan distribution, which is crucial for maintaining shape during growth [34]. Alongside this, we observed a significant increase in the level of elongation factor EF-Tu. In addition, its canonical function, EF-Tu, can be transported to the cell surface and participate in interactions with membrane receptors and the extracellular matrix on eukaryotic cell surfaces. Surface-localized EF-Tu has been shown to participate in adhesion to eukaryotic cells and bind complement factors and plasminogen [35,36]. EF-Tu also interacts with MreB in *E. coli* [37]. Since it is well known that eukaryotic EF-Tu (eEF1A) interacts with actin and influences cell shape, it has been hypothesized that a similar function may exist in prokaryotes. Indeed, EF-Tu has been shown to modulate MreB filament formation by binding MreB in a 1:1 ratio in *B. subtilis* and *E. coli* [38]. The increase in MreB and EF-Tu levels during growth on minimal media with PA may be related to the specific singularity of cell division under these growth conditions. PA induced a significant increase in the levels of chaperones IbpA and IbpB in both ZvL2-PA and MG1655-PA, which protect against irreversible protein aggregation during stress [39]. However, another interesting functional role of IbpA has been identified. It plays a role in FtsZ localization and chromosome segregation during cell division.

Among the common PA-induced proteins in ZvL2-PA and MG1655-PA that increase two-fold or more, we identified the serine/threonine protein kinase YeaG, whose function and targets are not well studied. For both isolates, the level of homocysteine methyltransferase MetE, involved in methionine biosynthesis, decreases.

Therefore, PA induced significant rearrangement in the abundance of membrane proteins in both the CD isolate ZvL2 and the laboratory strain MG1655. However, we were more interested in those proteins whose abundance changed only in the case of ZvL2, as these proteins may be associated with the formation of a virulent phenotype. We determined that PA induced a decrease in the abundance of the bifunctional protein TrpGD, involved in the biosynthesis of L-tryptophan, DNA protection during starvation protein Dps, and peptidoglycan-associated lipoprotein Pal in the CD isolate ZvL2. The latter is part of the Tol-Pal system, a protein complex of the outer membrane of Gram-negative bacteria that plays a role in the invagination of the outer membrane during cell division and is necessary for maintaining the integrity of the outer membrane. Disruption of the Tol-Pal complex leads to changes in the polar localization of chemoreceptors, cell motility, and chemotaxis. It is suggested that the Tol-Pal complex restricts the mobility of chemoreceptor clusters at cell poles and may be involved in regulatory mechanisms that coordinate cell division and chemosensory mechanism segregation [40]. Among the proteins whose abundance increased, we identified two enzymes involved in the biosynthesis of the 8-carbon sugar component of lipopolysaccharides—2-dehydro-3-deoxyphosphooctonate aldolase KdsA and arabinose 5-phosphate isomerase KdsD. 3-deoxy-D-manno-octulosonic acid (KDO) connects lipid A with the oligosaccharide chain and plays an important role in forming the barrier function and stability of the outer membrane [41]. It is known that the loss of LPS due to KDO deficiency destabilizes the membrane structure, presumably causing significant disruptions in membrane protein localization and cell division. In addition to these two proteins, the most noticeable differences were changes in the abundance of isoforms of the porin OmpA.

#### 2.3.2. Protein Modifications of OmpA Porin, Particularly Isoforms and Unique Amino Acid Substitutions, May Play a Role in ROS and Cytokine Activation upon ZvL2-PA Interaction with Neutrophils

One of the advantages of 2D electrophoresis is the identification of protein isoforms. The 2D map shows that the OmpA protein is represented in the form of isoforms that differ not only in pI value but also in molecular weight (Figure 2). Similar OmpA isoforms have been observed previously [42,43]. It is suggested that the formation of isoforms may be related either to sample preparation for 2D electrophoresis or due to post-translational modifications (PTM), amino acid substitutions, or deamidation. The molecular weight of OmpA in *E. coli* is 37428 Da, pI—5.65, which corresponds to spot No. 1 on the 2D map (Figure 2). Spots 2, 3, 4, and 5 are shifted to the acidic region. Spots 6–10 are located lower and differ in molecular weight by approximately 5 kDa. It has been shown that the electrophoretic mobility for fully denatured and native or partially denatured OmpA differs: OmpA is considered denatured when it forms a band around 35 kDa and “native” when it forms a band around 30 kDa [44]. In our case, the apparent molecular mass of fully denatured OmpA is 37.4 kDa. Spots 6-10 on the 2D map have an apparent molecular mass of about 32 kDa and, apparently, represent partially denatured OmpA. The relative abundance of OmpA isoforms varies greatly when comparing ZvL2-PA and ZvL2-GLU (Figure 2). PA induced an increase in the levels of some isoforms (spots 1-2, 6-7) and a significant decrease in others (spots 3-5 and 9). A completely different picture was observed in the case of the laboratory strain MG1655. It is characterized by the absence of two isoforms (spots 4 and 5) observed in ZvL2. PA caused a significant decrease in the levels of all OmpA isoforms except spot 1 (Figure 3). The quantitative assessment of the change in OmpA isoform levels for the CD isolate and the laboratory strain is shown in Figure 2 and Figure 3. The porin OmpA is a multifunctional protein that plays an important role in the pathogenesis process. The diversity of OmpA isoforms may be associated with its numerous functions. One of the reasons for the formation of isoforms is phosphorylation. Using 2D electrophoresis and staining with the Pro-Q Diamond dye specific for phosphate groups, we analyzed phosphorylated proteins in the membrane fraction isolated from ZvL2-PA cells (Figure 2D). The spots corresponding to isoforms 6–10 of the OmpA protein were stained with Pro-Q Diamond, indicating that they all carry phosphate groups. The shift of these spots to the acidic region may indicate that the number of phosphate groups increases from spot 6 to spot 10.

Although OmpA is a highly conserved protein, it has variable regions—four external loops connecting the membrane β-barrels of the N-terminal domain. These regions play a role in the protein’s interaction with the external environment. It has been suggested that variations in the sequences of certain genes (*fimH*, *chiA*, and *ompA*) may be associated with the virulent properties of AIEC [10]. Previously, we conducted whole-genome sequencing of the CD isolate ZvL2 [10]. We compared the amino acid sequences of OmpA from the laboratory strain K12 MG1655, the CD isolate ZvL2, and the reference AIEC LF-82 (Figure 4). We identified two amino acid substitutions (SNPs) in loop 4 (N176H and L186M) and two substitutions in the C-terminal domain (N228T and G276A) in the CD isolate ZvL2 compared to the laboratory strain and LF-82. Unlike LF-82, the K12 and ZvL2 isolates have a four-amino acid insertion (GASF) in loop 3. It has been shown that such an insertion is characteristic of invasive enterobacteria [45]. It cannot be excluded that the unique amino acid substitutions in ZvL2 may contribute to the diversity of OmpA’s functional properties.

#### 2.3.3. A Comparative LC-MS Analysis of Membrane Fractions Corroborated and Extended the Data Obtained through 2D-DIGE, Indicating That PA Induces Significant Changes in the Abundance of Membrane Proteins Compared to Glucose for Both the CD Isolate ZvL2 and the Laboratory Strain MG1655

We identified 382 differentially expressed proteins for the CD isolate ZvL2 and 284 for K12 MG1655 with a cutoff of FC = 1 and *p*-value ≤ 0.05 when comparing membrane fraction proteomes isolated from *E. coli* grown in M9 medium supplemented with PA versus glucose (Figure 5, Appendix A). Among these, 81 proteins exhibited increased abundance, and 28 proteins showed decreased abundance when the isolates were grown in medium with PA compared to glucose, common to both isolates (Figure 5C).

Figure 6 presents the functional groups of differentially expressed proteins. Transcription factors, lipoproteins, transporter proteins, and subunits of the NADH-dehydrogenase-cytochrome bo3 complex (NDC) were most notable among the upregulated proteins for both isolates. NDC facilitates the oxidation of NADH produced by the tricarboxylic acid cycle (TCA), generating a proton gradient for ATP synthesis (we also observed increased abundance of ATP synthase subunits for both isolates). This complex is crucial for maintaining a specific NADH/NAD+ ratio in the cell. An increase in reduced equivalents inactivates the TCA cycle, while an accumulation of oxidized equivalents activates it. NDC mediates the metabolic flux switch in bacteria. When PA is used as a carbon source, the TCA cycle is activated because the methyl citrate cycle of PA utilization produces pyruvate, which is further metabolized in the TCA cycle. Increased NDC components are associated with the metabolic switch to PA utilization in *E. coli*. Amino acid biosynthesis enzymes, transporter proteins, and proteins involved in translation were most prominent among the down-regulated proteins in both isolates. We previously demonstrated that *E. coli* cells grow significantly slower in M9 medium with PA compared to glucose. The reduction in the abundance of these proteins is associated with the slowed growth of our isolates in PA-containing medium. Lipid metabolism proteins and proteins related to the transport and biosynthesis of LPS were identified among the upregulated proteins for the CD isolate ZvL2, distinguishing it from the laboratory strain. Amino acid biosynthesis enzymes were among the down-regulated proteins. For the laboratory strain, a large number of ribosomal proteins were found among the down-regulated proteins, which were not detected in the CD isolate.

We were interested in tracking changes common to both strains and those specific to the CD isolate but not observed in the laboratory strain. We identified several main directions in protein abundance changes when the isolates were grown in PA medium compared to glucose.

##### PA Upregulates Envelope Stress Proteins in Both Strains

We noted that both strains exhibit an increase in envelope stress-related proteins when cultured in PA medium compared to glucose (Figure 7A). We observed increased abundance of sensor lipoproteins within two-component systems. Among them is the OmpR protein, a member of the EnvZ/OmpR two-component regulatory system involved in osmoregulation. It is known to regulate the expression of outer membrane proteins OmpF and OmpC in response to changes in medium osmolarity. The total amount of OmpF and OmpC in *E. coli* cells remains relatively constant, but their levels are regulated in response to environmental stimuli [46]. Additionally, OmpR is known to regulate a broad range of genes, including the major flagellar regulon flhDC in *E. coli* [47], the csgDEFG operon, and curli fimbriae production [48,49,50]. The lipoprotein NlpE was also included. NlpE is known to be a universal envelope sensor that uses its structure, localization, and interactions with other envelope proteins to initiate adaptation to various signals [51]. Its C-terminal domain is involved in environmental sensing, while the N-terminal domain interacts with the sensor kinase CpxA, activating the Cpx regulatory pathway. NlpE also regulates the interplay between two crucial envelope biogenesis processes: lipoprotein sorting and protein folding [52]. We also found that PA induces an increase in α-trehalose-phosphate synthase OtsA and sodium/proline symporter PutP levels, which are associated with cellular responses to osmotic stress. Thus, PA activates two component sensor regulatory systems related to osmotic stress in both strains. Among envelope stress-related proteins, we found that only in the case of ZvL2-PA does the abundance of the PhoP protein, a member of the PhoP/PhoQ two-component regulatory system (involved in adaptation to low Mg2+ environments, control of acid resistance genes, and antimicrobial peptide resistance), the sensor histidine kinase KdpD (part of the KdpE/KdpD two-component regulatory system responsible for osmotic stress response, K+ limitation, and reduced cell turgor), and the mechanosensitive channel protein MscL increase. PhoPQ controls the transport of cardiolipin to the outer membrane, with elevated levels found in bacteria under stress, particularly osmotic pressure [53,54]. We found increased levels of two cardiolipin biosynthesis enzymes, cardiolipin synthases (ClsB and ClsC), in ZvL2-PA.

##### PA-Induced Changes in Regulator Protein Abundance Differ between CD Isolate and Laboratory Strain

We identified several regulatory proteins that were upregulated in ZvL2-PA (Figure 7B). These include the Rcs signaling system component and the outer membrane sensor lipoprotein RcsF. The Rcs signaling system regulates the transcription of numerous genes, including those involved in colanic acid synthesis, biofilm formation, cell division, and outer membrane protein synthesis [55]. RcsF is exported to the outer membrane, presumably through the Lol system [56]. Another regulator protein was ArcA, a member of the ArcB/ArcA two-component regulatory system. Recent data on ArcAB function under aerobic conditions challenge its canonical role as a strictly anaerobic global regulator. In addition to its role, ArcAB connects the electron transport chain with gene expression and is essential for bacterial resistance to reactive oxygen species under aerobic conditions, possibly due to its impact on bacterial metabolism. ArcA regulates gene expression and helps control levels of glutathione, NADH, intracellular ROS, and the pyruvate dehydrogenase complex, thereby modulating the cell’s redox potential [57]. ArcA modulates bacterial adaptive responses to stress within phagocytes and regulates porin gene expression in response to stress factors during infection. It controls at least 100 operons involved in the TCA cycle and energy metabolism [58,59].

Both isolates showed increased abundance of the global transcription regulator Cra, which plays a crucial role in carbon metabolism regulation, the regulator MtfA involved in the modulation of the glucose-phosphotransferase system, and the cysteine regulon gene regulator CysB. Cra activates the transcription of genes encoding enzymes in the TCA, glyoxylate shunt, and gluconeogenesis and represses genes involved in sugar catabolism, which is consistent with the switch to PA growth. In MG1655-PA, we observed an increased level of GlcC, the transcriptional activator of the glcDEFGB operon associated with glycolate utilization. In ZvL2-PA, the levels of transcription factors Crl and MlrA, which activate the csgBAC operon and the master regulator of biofilm formation, part of the curli biosynthesis signaling cascade, were increased. Additionally, only in ZvL2-PA did we observe an increase in the TreR repressor of the treBC operon. This increase is linked to trehalose regulation in response to high osmolarity. Under high osmolarity, trehalose synthesis increases as a protective mechanism, while trehalose-6-phosphate transport and its conversion to glucose and glucose-6-phosphate, regulated by treBC genes, decrease. The increased abundance of the TreR repressor of these two genes is related to regulation in response to high osmolarity.

##### PA Upregulates LPS Proteins Only in the CD Isolate ZvL2

LPS biosynthesis, transport, and modification were the critical trends in the proteome change when switching the carbon source from glucose to PA. Switching to PA resulted in an increase in the content of proteins involved in these processes, mainly in ZvL2-PA, whereas in MG1655-PA their content remained unchanged or decreased (Figure 7C). These included LPS biosynthesis enzymes (LpxACM, GalFU) and LPS transport proteins to the outer membrane (LptABDEG). The outer membrane of many gram-negative bacteria, such as *E. coli*, *S. enterica*, *S. typhimurium*, and *K. pneumoniae*, modifies its LPS with additional functional groups such as fatty acyl chains, phosphoethanolamine, 4-amino-4-deoxy-L-arabinose, and glycosyl groups to enhance resistance to antimicrobial peptides [60]. The ArnA protein was identified among the unique proteins in the ZvL2-PA. It catalyzes the transfer of 4-amino-4-deoxy-L-arabinose from UDP to undecaprenyl phosphate. The modified arabinose binds to lipid A, allowing *E. coli* to exhibit resistance to polymyxin and cationic antimicrobial peptides [61]. It is known that PhoPQ (whose level also increases in ZvL2-PA) binds to the promoter of the *arnB* gene and directly regulates the biosynthesis of 4-amino-4-deoxy-1-arabinose, its covalent binding to lipid A on the inner membrane, and its transport to the outer leaflet of the outer membrane [62]. However, in both isolates, the level of phosphoethanolamine transferase EptC, which catalyzes the attachment of the phosphoethanolamine fragment to the lipopolysaccharide core of the outer membrane and thus plays a role in lipid A modification, decreased. In the case of ZvL2-PA, we observed an increase in the abundance of the MlaA protein. It is involved in the ATP-dependent transport of mislocalized lipids from the outer leaflet (OM) to the inner leaflet (IM) [63]. The inner membrane protein YejM, with five predicted transmembrane domains, was another protein whose level increases in the ZvL2-PA. YejM plays a crucial role in maintaining LpxC levels and thus regulates the optimal phospholipid/LPS ratio in the outer membrane, which is vital for cell survival. It has been shown that the deletion of the five predicted transmembrane domains of YejM is lethal for Gram-negative bacteria [63]. In *E. coli* YejMPD knockout strains, the synthesis and delivery of cardiolipin, necessary for the outer membrane structure, are disrupted. Knockout strains are characterized by increased membrane permeability and activation of PhoPQ two-component system genes, and such bacteria do not survive inside host cells [64].

##### PA Causes an Increase in the Abundance of Porin OmpA Only for the CD ZvL2 Isolate

The PA-induced increase in the abundance of outer membrane proteins OmpF, SlyB, TolC, LoiP, and NlpE was observed for both strains (Figure 8A). The abundance of outer membrane proteins OmpA, Blc, and LolB increased in the PA medium in the ZvL2-PA and decreased or remained unchanged in the MG1655-PA. Lipocalin Blc is involved in the transport of lipids necessary for maintaining membranes under stress conditions. LolB plays a critical role in the incorporation of lipoproteins into the outer membrane after they are released by the LolA protein.

##### PA Induces Changes in the Abundance of Transport and Iron Retention Proteins for Both the CD Isolate ZvL2 and the Laboratory Strain MG1655

An increase in the abundance of amino acid transport proteins, as well as the pyruvate/proton symporter CstA and the nucleoside transporter Tsx, was observed for both MG1655-PA and ZvL2-PA. Activation of amino acid transport under the growth conditions of isolates in minimal medium supplemented with PA may be due to starvation compared to growth on glucose (Figure 8B). An increase in the level of proteins related to iron transport and retention was observed for both isolates during growth in minimal medium with PA. Among them are ferrienterobactin transporters FepABD, ferrichrome porin FhuA, catecholate siderophore receptor CirA, enterobactin synthetase component EntB, and enterobactin esterase Fes (Figure 8C).

Previously, we isolated *E. coli* strains from biopsies and the lumen of patients with Crohn’s disease capable of attaching to and invading eukaryotic cells, surviving, and replicating within macrophages. We showed that if such isolates were passaged in minimal medium M9 with PA, they significantly increased these virulent properties. If they were passaged in minimal medium with glucose, they lost them [23]. This phenotypic switch effect was reversible and was not observed for the laboratory strain *E. coli* MG1655. We hypothesized that comparing the virulent (ZvL2-PA) and non-virulent (ZvL2-GLU) phenotypes of the CD isolate and using the laboratory strain as a control would allow us to identify factors responsible for the formation of the virulent phenotype. To summarize, in this study we showed that the intensity of the respiratory burst of neutrophils, assessed by the production of superoxide anion radical, increased when interacting with ZvL2-PA neutrophils, unlike ZvL2-GLU and the laboratory strain. We also found that neutrophil activation required opsonization of the bacteria, and ROS production activity increased significantly in the presence of serum. The high intensity of singlet oxygen release by neutrophils is accompanied by active oxygen consumption and can lead to local hypoxia and increased expression of the hypoxia factor HIF-1 alpha. Experiments with transgenic mice capable of expressing CEACAM6 showed that AIEC (unlike *E. coli* K12) induces increased expression of HIF-1 alpha, which is accompanied by enhanced CEACAM6 expression. In turn, the interaction of AIEC with CEACAM6 through FimH (an adhesin located at the tip of type 1 pili) promotes successful colonization and invasive capability of AIEC [65]. In our earlier work, we showed that AIEC can evade neutrophil action by binding mucin and thus shielding molecular epitopes that activate neutrophils [36]. At the same time, neutrophil-induced microenvironmental hypoxia and HIF stabilization may promote CEACAM expression by epithelial cells and AIEC adhesion through CEACAM6-FimH interaction. Differences in NADPH oxidase activation and ζ-potential for ZvL2-GLU and ZvL2-PA indicate changes in the envelope cell, including changes in the composition and structure of LPS, as well as the abundance of membrane proteins. In this study, we focused on changes in the proteomes of enriched membrane fractions isolated from laboratory strains and CD-isolated cells grown in minimal medium with glucose or PA. Comparative proteomic analysis showed that PA causes significant changes in the abundance of membrane proteins in both strains. Both strains experience envelope stress. Both 2D electrophoresis and LC-MS analysis revealed a considerable number of differentially expressed proteins common to the CD isolate and the laboratory strain. In both cases, PA increases the level of sensory proteins in two-component systems (EnvZ/OmpR), α-trehalose phosphate synthase OstA, and sodium/proline symporter PutP, associated with cellular response to osmotic stress. The level of the universal envelope sensor NlpE, which interacts with other proteins to adapt to various external signals, also increases. Only in ZvL2-PA did we find an increase in the abundance of the RcsF receptor, which triggers the Rcs regulatory system, one of the main two-component systems protecting cells from envelope stress. The outer membrane protein assembly complex BAM assembles a complex of porins (OmpA, OmpC, and OmpF) with the RcsF receptor. As a result of stress and perturbations in LPS, the Omp-RcsF complex breaks down, releasing RcsF, which triggers Rcs signaling [66]. Konovalova et al. showed that the electrostatic charge interaction between LPS and RcsF plays an important role in sensing external signals [67]. It has also been shown that Rcs activation results in mucoid (resulting from colanic acid production) and non-motile (inhibition of flhDC flagellar operon genes) *E. coli* cells [68]. Activation of RCs signaling in ZvL2-PA can lead to colanic acid production, which can be a protective mechanism under stress and survival conditions in the host. Additionally, only in the CD isolate did we find increased abundance of PhoP. It has been shown for *S. Typhimurium* that PhoPQ regulates the activity of genes encoding proteins and enzymes that function to increase OM-lipid hydrophobicity, a property that prevents cationic antimicrobial peptides from binding and inserting into the membrane to ultimately kill the microbe [69,70]. The PhoPQ regulators coordinate chemical remodeling of lipopolysaccharides, alterations in the levels of OM proteins, and palmitoylation of lipid A and phosphatidylglycerols within the OM outer leaflet.

Switching the carbon source from glucose to PA entails changes in cellular metabolism, which is always associated with disruption of the NAD+/NADH balance and changes in the cell’s redox state. An increase in the level of ArcA (the ArcB/ArcA regulatory system), which controls these processes, was observed only in the case of ZvL2-PA [57].

A particularly interesting group of proteins for us were those involved in LPS synthesis, transport, and modification. We observed an increase in these proteins in ZvL2-PA and a decrease in MG1655-PA. This was the most pronounced difference in membrane proteomes between the CD isolate and the laboratory strain. LPS is a major outer membrane component, serving as a barrier against antimicrobial agents and a key virulence factor [71]. The properties of LPS depend on its chemical nature, acetylation, phosphorylation, and various modifications of lipid A (such as phosphoethanolamine, 4-amino-4-deoxy-L-arabinose, hexosamines, etc.). Among the unique proteins in ZvL2-PA, we identified ArnA, which catalyzes the transfer of 4-amino-4-deoxy-L-arabinose from UDP to undecaprenyl phosphate, participating in lipid A modification. Propionyl-CoA, formed during the methylcitrate cycle when PA is utilized, might also participate in outer membrane modifications, providing protection against the toxic excess of propionyl-CoA [72]. The increase in the level of LPS biosynthesis enzymes, LPS transporters, and some factors that promote LPS modification leads to the conclusion that changes in the LPS structure are important for the formation of the virulent phenotype of ZvL2. These may be associated with either an increase in LPS mass or modifications of lipid A or polysaccharide chains. Activation of Rcs signaling and PhoP/Q in ZvL2-PA may be directly related to these changes. They may be an additional stimulus that provides protection to the CB isolate under stress conditions. The role of LPS synthesis proteins in the intracellular persistence of AIEC in dendritic cells and activation of IL-23 secretion by them has already been shown in the work of G. Leccese et al. It was shown that a knockout mutant for the rfaP gene induced a sharp decrease in both the intracellular persistence of AIEC and the secretion of IL-23 in dendritic cells, which inhibited the shift of cTh17 cells towards the pathogenic pTh17 phenotype without affecting the viability and function of protective Th17 cells [4].

Another interesting group of changing proteins are porins. Increased levels of porin OmpF were characteristic of both strains, regulated in response to environmental stimuli, particularly by the EnvZ/OmpR system in response to osmolarity changes. However, we found differences between the CD isolate and the laboratory strain regarding the most abundant porin in *E. coli*, OmpA. Using 2D electrophoresis, we found that OmpA forms multiple isoforms, possibly related to phosphorylation. Comparing proteomic 2D maps of ZvL2 and MG1655, we found that PA decreases most OmpA isoforms in MG1655-PA, while ZvL2-PA shows a different pattern: some isoforms decrease while others increase significantly (Figure 2). OmpA is the main protein of the outer membrane of *E. coli* and is highly conserved among a wide range of Gram-negative bacteria. In addition to its main function, OmpA is involved in the structuring and stability of the outer membrane of bacteria under stress [73]. In the work with the reference CD isolate LF82, it was shown that OmpA binds to the Gp96 protein, which leads to invasion of *E. coli* into epithelial cells [16]. It also blocks complement activation by binding to C4-binding protein [74], induces the anti-apoptotic factor Bclxl in macrophages, preventing apoptosis [75], and plays a role in biofilm formation in *E. coli* [76]. OmpA is a major target for neutrophil elastase (NE). *E. coli* secretes large amounts of OmpA to bind NE during neutrophil interaction, avoiding damage [45,77]. OmpA’s multifunctionality might be due to post-translational modifications such as phosphorylation. In our proteome analysis, PA significantly increased the abundance of serine/threonine protein kinase YeaG, whose functions and targets are not well studied. YeaG’s association with the membrane fraction suggests it might phosphorylate membrane proteins, including OmpA. YeaG is known to regulate metabolic fluxes during the transition from glucose to alternative carbon sources such as malate [78]. Isocitrate lyase AceA was identified as a YeaG substrate, phosphorylated only in the presence of malate. In our case, YeaG may play a similar role in switching the growth of our isolates from glucose to PA as a carbon source. Another post-translational modification of OmpA under growth with PA could be propionylation of lysine. Propionyl-CoA, formed during PA utilization, is a source of protein post-translational modifications, with a broad range of substrates identified in *E. coli*, potentially regulating metabolic processes [79]. We also identified two SNPs in OmpA’s variable loop 4 (N176H and L186M) and two substitutions in the C-terminal domain (N228T and G276A), which might be functionally significant and adaptation in host.

Thus, the difference between virulent ZvL2-PA and non-virulent ZvL2-GLU may be related to the activation of Rcs and PhoP/Q signals, which promote the formation of defense mechanisms, leading to the formation of biofilms as well as capsules. PA also induces changes in the composition and/or modification of LPS. Specific OmpA isoforms might play a role in forming the virulent ZvL2-PA phenotype and participate in neutrophil NADH oxidase activation.

## 3. Materials and Methods

### 3.1. Cell Cultures

*E. coli* isolate ZvL2 (CD isolate ZvL2) was obtained by an endoscopic surgery of a patient with Crohn’s disease at the Loginov Moscow Clinical Scientific Center. Sample collection was carried out in accordance with the requirements of the local ethics committee based on the informed consent of the patient. CD isolate ZvL2 was sequenced by our group [8] and identified as belonging to phylogroup A. CD isolate ZvL2 and laboratory strain K-12 MG1655 and were cultivated on M9 medium supplemented with 50 mM glucose (ZvL2-GLU and MG1655-GLU) or 20 mM sodium propionate (ZvL2-PA and MG1655-PA) at 37 °C and 180 rpm. The growth curve was monitored using OD at 600 nm. Cultures in the early logarithmic phase were used for experiments. Blood samples were withdrawn from healthy volunteers with their informed consent, in accordance with the protocol approved by the ethical committee of Lopukhin Federal Research and Clinical Center of Physical-Chemical Medicine (protocol No. 2022/10/05). The *E. coli* ZvL2 from the CD patient was previously used by us in a study published in 2017 (doi: 10.1186/s12864-017-3917-x.). This study involving *E. coli* samples from the CD patient was approved by the ethics committees of the Central Research Institute of Gastroenterology and the State Research Center of Proctology.

### 3.2. Luc-CL and Cytokines

Luc-CL was measured with the use of the luminometer Lum1200 (DiSoft, Moscow) in 0.5 mL of Krebs–Ringer solution (pH 7.4) with 0.1 mM lucigenin (Luc-CL), 2% of autologous blood serum or without serum as indicated in the text, neutrophils 0.5–0.7 × 10^6^ cells mL^−1^, and *E. coli* with a final concentration of 2.5 × 10^8^ CFU/mL. Spontaneous CL was measured before *E. coli* addition, then the bacterial sample was added and CL was registered until maximum values were reached; the CL amplitude (V) was calculated as the difference between maximum and spontaneous values; the integral CL values were also calculated. 0.15 M NaCl was added instead of bacteria to blank control probes. Supernatants obtained after the interaction of *E. coli* strains with neutrophils were used to measure the content of cytokines. The assay of TNF-alpha, IL-6, and IL-1 beta was carried out using commercial test systems for ELISA (“Cytokine”, Russia) according to the manufacturer’s instructions.

### 3.3. ζ-Potential Measurement

The cell culture was grown to the middle of the logarithmic phase. The cell culture was diluted mQ to 10^7^ to 10^8^ CFU mL^−1^ cell count before measurement. The ζ-potentials of bacteria were measured using Zetasizer (Nano ZS, Malvern, UK).

### 3.4. Isolation of Membrane Fraction

The cells of the CD isolate and laboratory strain K12 MG1655 were centrifuged for 5 min at 4 °C, 5000× *g*. The cell pellets were resuspended in 2 mL of 10 mM Tris (pH 8.0) and 20 μL of lysozyme (stock 10 mg/mL) and were incubated at 4 °C for 2 min. An equal volume (2 mL) of 0.3 mM EDTA was added dropwise, constantly stirring on a magnetic stirrer (Biosan, Riga, Latvia) on ice. An amount of 4 µL of protease inhibitors and 4 µL of 1 M DTT were added and incubated at 4 °C for 30 min. Then, OD at 450 nm was measured. The cells were disrupted with an ultrasonic disintegrator (Branson, MO, USA) until the OD value at 450 nm has decreased to approximately 50% of its original value. The unbroken cells were removed by centrifugation for 20 min at 4 °C, 3000× *g*. The supernatant was collected to ultracentrifugation for 90 min at 4 °C, 100,000× *g* in a 31 Optima L-90K ultracentrifuge (Beckman Coulter Inc., Brea, CA, USA); the pellets were resuspended in 1 mL of 10 mM Tris (pH 8.0) with 0.25 M sucrose and ultracentrifuged in the same conditions.

### 3.5. 2D-DIGE

2D-DIGE was performed using the method we described earlier [21]. Before electrophoresis, membrane fractions were resuspended in a 40 mM Tris-HCl buffer (pH 9.5) with 8 M urea, 2 M thiourea, 4% CHAPS, 2% (*w/v*) NP-40, and 2% Ampholytes (pH 4–6 and pH 3.5–10). Protein concentration in the supernatant was measured by the Bradford method using Quick Start Bradford dye (BioRad, Hercules, CA, USA). The sample proteins were labeled with Cy3 (green) or Cy5 (red) CyDye DIGE Fluor minimal dyes (Lumiprob, Moscow, Russia) according to the manufacturer’s instruction (400 pmol for 50 μg protein). After electrophoresis, the gel was scanned on the scanner TyphoonTrio (Amersham) at a laser wavelength of 532 nm (green fluorescence) and 633 nm (red fluorescence). Protein spot quantitation was performed using PDQest 8.0 software. Then, the gels were stained with silver as described in [80].

### 3.6. Tryptic Digestion and Protein Identification

The protein spots (gel pieces) were excised and washed with a mixing solution containing 15 mM tetrathionate and 50 mM potassium ferrocyanide to remove silver. Then, the gel pieces were washed in mQ water and dried in 100% acetonitrile. An amount of 3 µL of trypsin solution (40 mM ammonium bicarbonate, 10% acetonitrile, and 40 nM trypsin) was added to the gel pieces, and their incubation for 30 min on ice and then for 16 h at 37 °C was performed. To the gel pieces were added 6 µL of 0.5% *v/v* TFA in mQ and incubated in an ultrasonic bath for 10 min and then for 1 h at room temperature. Mass spectrometric analysis was performed on an Ultraflex II MALDI-ToF-ToF (Bruker Daltonics). For MALDI mass spectrometry analysis of tryptic peptides from 2D gel, 20 mg/mL of 2,5-Dihydroxybezoic Acid (2,5-DHB) in 50% water/acetonitrile and 0.1% TFA was used. MALDI-TOF spectra were acquired using Ultraflex II MALDI-TOF (Bruker, Billerica, MA, USA) for positive ions in reflectron mode over the range 800–4000 Da. Each spectrum was accumulated using 2000 laser shots (Nd-YAG, 100 Hz, λ = 355 nm). The peak list was filtered to remove trypsin and human keratin masses that often were observed due to some degree of contamination. The searches in MASCOT (version 2.5.1, Matrix Science Ltd., London, UK) (80 ppm accuracy between experimental and theoretical values) were performed with carbamidomethylation of the cysteine residues as fixed modification and the oxidation of methionines as variable. Proteins were identified against the protein UniProt database of reference proteomes for K12 *E. coli* (https://www.uniprot.org/proteomes/UP000000625, accessed on 5 September 1997). The identification cutoff was 44 (*p* < 0.05).

### 3.7. Fluorescent Staining of Phosphorylated Proteins

Two-dimension gel electrophoresis (2D-DIGE) of membrane proteins of the ZVL2 strain was performed as described above. All actions were carried out on an orbital shaker. The gels were fixed in 50% methanol and 10% acetic acid for 30 min and then washed with water 3 times for 15 min. The gels were stained with Pro-Q^®^ Diamond Stain (Invitrogen, Waltham, MA, USA) for 2 h. Dye was stopped in 20% acetonitrile and 50 mM sodium acetate (pH 4) 3 times for 30 min. The gels were washed in ultrapure water twice for 5 min. Poststaining images of the gels were obtained by scanning the gels with the scanner TyphoonTrio (Amersham, UK) at 550/580 nm.

### 3.8. Free-Gel Digestion of Protein of Membrane Fractions

An amount of 10 μL of 10% surfactant RapiGest (Waters) was added to the pellets of membrane fraction. After incubation at 4 °C for 30 min, the samples were resuspended in 100 µL of 100 mM Tris-HCl buffer (pH 8.5) containing 0.1% RapiGest. After incubation for 20 min, samples were centrifuged at 14,000× *g* at 4 °C for 10. The supernatant was collected, and the protein concentration was measured using a BCA Assay Kit (Sigma, Tokyo, Japan). Disulfide bonds were reduced in the supernatant containing 200 μg of total protein by the addition of Tris(2-carboxyethyl) phosphine hydrochloride (TCEP) (Sigma) to a final concentration of 5 mM, and the reaction was incubated at 37 °C for 60 min. To alkylate-free cysteines, chloroacetamide (BioRad) was added to a final concentration of 30 mM, and the solution was placed at room temperature (RT) in the dark for 30 min. Trypsin (Trypsin Gold, Mass Spectrometry Grade, Promega, Tokyo, Japan) to achieve a final trypsin: protein ratio of 1:50 (*w/w*) was added, and samples were incubated at 37 °C overnight. To stop trypsinolysis and to degrade the acid-labile RapiGest, trifluoroacetic acid (TFA) was added to a final concentration of 0.5% (*v/v*) (the pH should be less than 2.0), and the samples were incubated at 37 °C for 45 min. Further, the samples were centrifuged at 14,000× *g* for 10 min to remove RapiGest. The peptide extract was desalted using a Discovery DSC-18 tube (Supelco, St. Louis, MO, USA) according to the manufacturer’s protocol. Peptides were eluted with 1 mL of 75% acetonitrile (ACN) solution containing 0.1% TFA, dried in a SpeedVac (Labconco, Kansas City, MO, USA), and resuspended in a 3% ACN solution containing 0.1% TFA to a final concentration of 5 μg/μL.

### 3.9. LC-MS Analysis

The LC-MS analysis of proteome samples was performed using an Orbitrap Q Exactive HF-X (Thermo Fisher Scientific, Waltham, MA, USA) mass spectrometer. For ionization, a nano-electrospray (nano-ESI) source was used in conjunction with high-pressure nanoflow chromatography UPLC Ultimate 3000 (Thermo Fisher Scientific, Waltham, MA, USA). The lab-made reverse-phase column (ID 100 mm with length 500 mm of fused silica TSP100375 (Molex, Lisle, IL, USA) was packed with phase Kinetex C18, 2.4 μm (Phenomenex, Torrance, CA, USA) using a pressure injection cell (Next Advance, Troy, NY, USA). During the HPLC run, it was thermostatically controlled at 60 °C. Samples were loaded in buffer A (0.1% formic acid) and eluted with a linear (90 min) gradient of 3 to 55% buffer B (0.1% formic acid, 80% acetonitrile) at a flow rate of 220 nl/min. Mass spectrometric data were stored during automatic switching between MS1 scans and up to 12 MS/MS scans (TopN method). The target AGC value for MS1 scanning was set to 3 × 10^6^ in the range 390–1400 m/z with a maximum ion injection time of 45 ms and resolution of 60,000. The precursor ions were isolated at a window width of 2.0 m/z and then were fragmented by high-energy dissociation with a normalized collision energy of 30 eV. MS/MS scans were saved with a resolution of 30,000 at 400 m/z and AGC of 1 × 10^5^ for target ions with a maximum ion injection time of 50 ms. The obtained proteomic data were uploaded in PRIDE (project accession: PXD054715, project DOI: 10.6019/PXD054715).

### 3.10. Protein Identification, Quantification, and Comparative Proteomic Profiling

The Nextflow pipeline QuatMS (10.5281/zenodo.7754148) with the DDA-LFQ analysis method was used to process mass spectrometric proteomic data—quantification and identification of proteins. A file with protein sequences for the collected and annotated genomes of MG1655 and ZvL2 *Escherichia coli* strains was taken as a protein database. The R programming language and the functionality of the MSstats library were used for postprocessing the file with intensities, detected peptides—normalization and statistical analysis.

### 3.11. Data Processing and Statistical Analysis

PDQuest 8.0 software was used to analyze proteomic data using 2D-DIGE. The statistical processing of the results with neutrophils was performed using Excel and Statistica 12.0 programs. The results are presented as mean ± SD, where the mean is the arithmetic mean and SD is the standard deviation. Statistical significance of differences between groups was assessed using the Mann–Whitney test.

## 4. Conclusions

We showed that the intensity of the respiratory burst of neutrophils, assessed by the production of superoxide anion radical, increased when interacting with ZvL2-PA, unlike ZvL2-GLU and the laboratory strain. The activation of neutrophils required opsonization of the bacteria, and the activity of ROS production significantly increased in the presence of serum. Differences in NADPH oxidase activation and ζ-potential for ZvL2-GLU and ZvL2-PA may be associated with the activation of Rcs and PhoP/Q signaling pathways, which promote the formation of protective mechanisms, leading to the formation of biofilms, capsules, and changes in the composition and/or modification of LPS. Certain isoforms of OmpA may play a role in the formation of the virulent phenotype of ZvL2-PA and participate in the activation of NADPH oxidase in neutrophils.

The limitations of our study: We limited our study to only one AIEC isolate, which limits the generalizability of the results and does not yet provide evidence that the observed effects would be the same for other AIEC isolates.

## Figures and Tables

**Figure 1 ijms-25-10118-f001:**
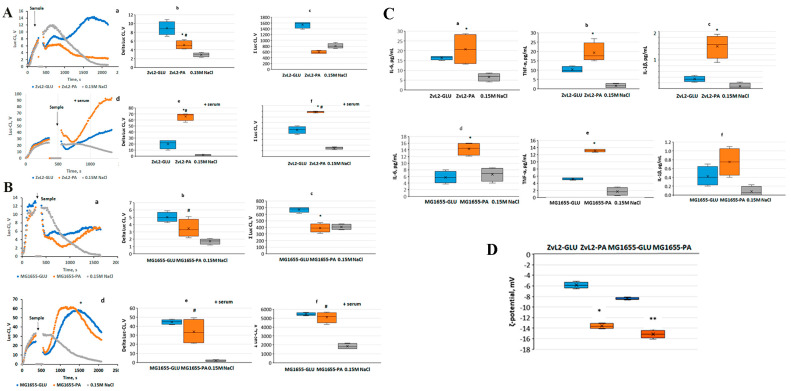
Time-course of neutrophil Luc-enhanced CL response, stimulated with ZvL2-GLU, ZvL2-PA (**A**), MG1655-GLU, MG1655-PA (**B**), or 0.15M NaCl in absence (a–c) or in the presence (d–f) of 2% autologous serum. b, e—amplitude (delta) and c, f—integral intensity (∑) of neutrophil Luc-enhanced CL response. (**C**)—cytokines extracellular content (pg/mL) after incubation of neutrophils with *E. coli*. (**D**)—ζ -potential measured in cell culture ZvL2-GLU, ZvL2-PA, and MG1655-GLU, MG1655-PA (^#^ *p* ˂ 0.05 vs. 0.15 M NaCl; * *p* ˂ 0.05 vs. ZvL2-GLU; ** *p* ˂ 0.01 according to Mann–Whitney test).

**Figure 2 ijms-25-10118-f002:**
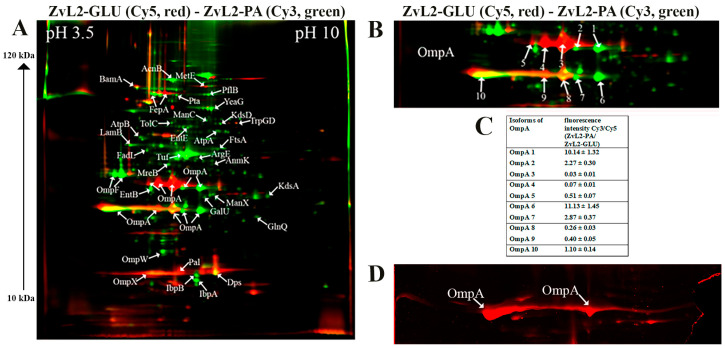
(**A**)—comparative proteomic analysis of membrane fractions of CD isolate ZvL2 grown on glucose (GLU) and propionate (PA) for 5 passages revealed by differential 2D gel electrophoresis (pH range 3.5–10). Membrane proteins isolated from ZvL2-PA were stained with green cyanine, and membrane proteins isolated from ZvL2-GLU were stained with red cyanine. Green spots on the 2D map correspond to membrane proteins, the level of which increases by 2 or more times, and red spots—decrease by 2 or more times when comparing isolates grown on PA relative to GLU. The cutoff score for protein identification in the mascot engine was 44 (*p* < 0.05). (**B**)—a fragment of a 2D map with the location of spots corresponding to the OmpA. (**C**)—an increase or decrease in the level of OmpA isoforms in the ZvL2 isolate grown on the M9 medium supplemented with PA or glycose. (**D**)—the spots corresponding to isoforms 6–10 of the OmpA protein stained with Pro-Q Diamond.

**Figure 3 ijms-25-10118-f003:**
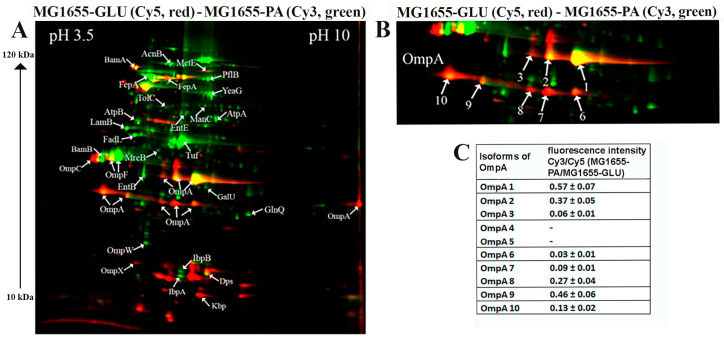
(**A**)—comparative proteomic analysis of membrane fractions of K12 MG1655 grown on glucose (GLU) and propionate (PA) for 5 passages revealed by differential 2D gel electrophoresis (pH range 3.5–10). Membrane proteins isolated from MG1655-PA were stained with green cyanine, and membrane proteins isolated from MG1655-GLU were stained with red cyanine. Green spots on the 2D map correspond to membrane proteins, the level of which increases by 2 or more times, and red spots—decrease by 2 or more times when comparing isolates grown on PA relative to glucose. The cutoff score for protein identification in the mascot engine was 44 (*p* < 0.05). (**B**)—a fragment of a 2D map with the location of spots corresponding to the OmpA. (**C**)—an increase or decrease in the level of OmpA isoforms in the MG1655 grown on the M9 medium supplemented with PA or glycose.

**Figure 4 ijms-25-10118-f004:**
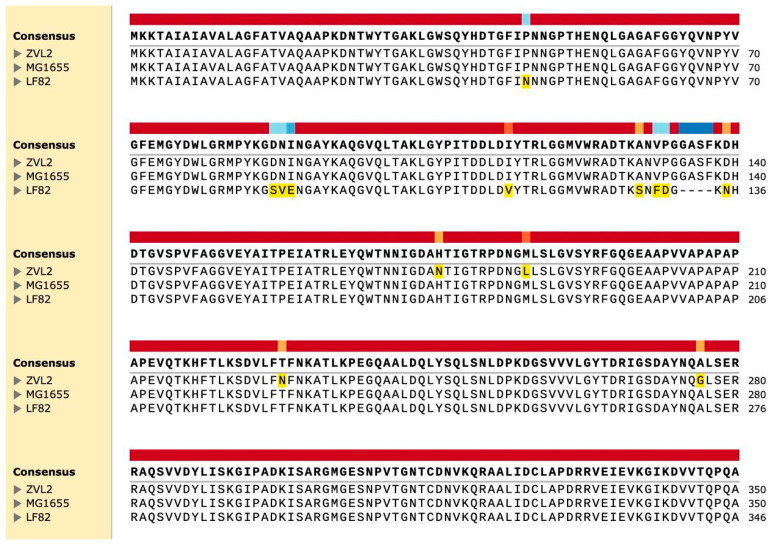
Comparative analysis of amino acid sequences of the OmpA protein of CD ZvL2 isolate, laboratory strain K12 MG1655, and reference strain AIEC LF-82.

**Figure 5 ijms-25-10118-f005:**
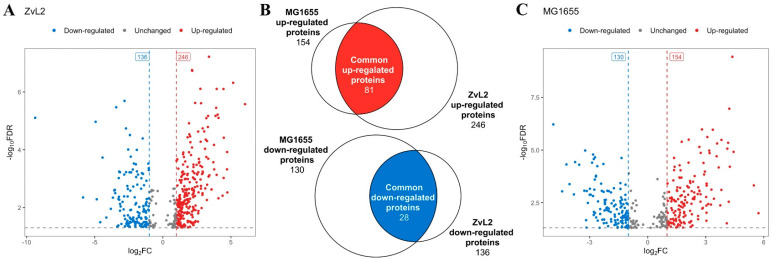
Vulcano plots indicating the number of differentially expressed proteins in CD isolates ZvL2 (**A**) and laboratory strain MG1655 (**C**), grown on M9 medium supplemented with sodium propionate (PA) relative to glucose. (**B**)–Venn diagrams showing the number of common and unique differentially expressed proteins for each isolate.

**Figure 6 ijms-25-10118-f006:**
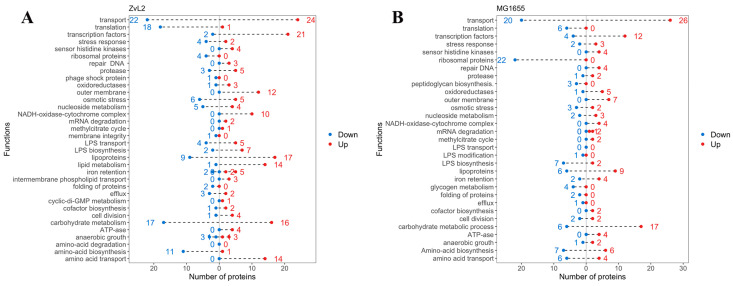
The functions of proteins, the abundance of which increased (right, red) or decreased (left, blue) in ZvL2-PA compared to ZvL2-GLU (**A**) and MG1655-PA compared to MG1655-GLU (**B**).

**Figure 7 ijms-25-10118-f007:**
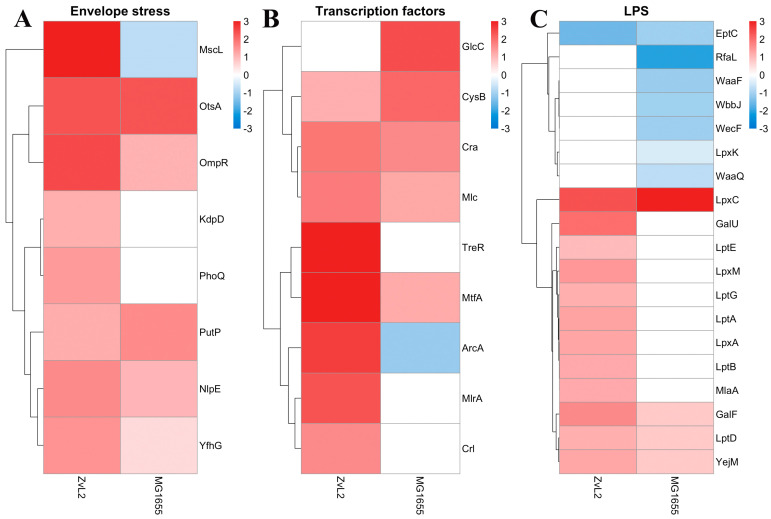
Differentially expressed envelope stress proteins (**A**), transcription factors (**B**), and LPS transport and LPS biosynthesis proteins (**C**) in ZvL2-PA compared to ZvL2-GLU and MG1655-PA compared to MG1655-GLU. Heatmaps represent -log2 Fold Change (FC) values (*p* value ≤ 0.05). White boxes are the proteins with non–significant changes.

**Figure 8 ijms-25-10118-f008:**
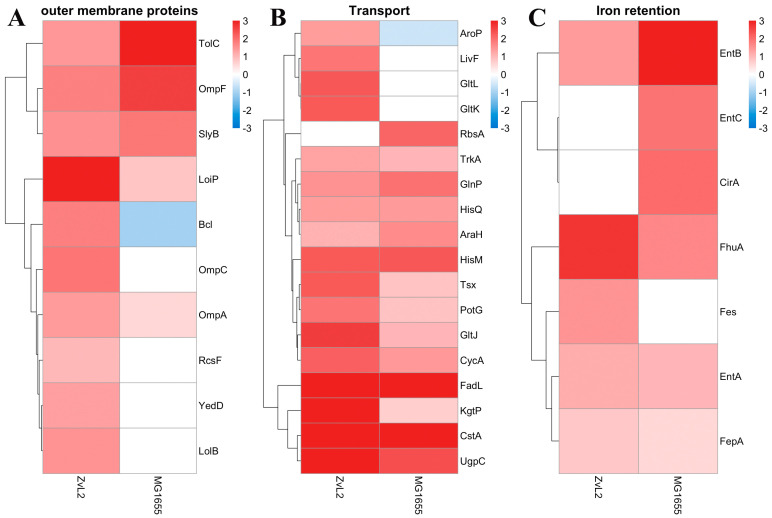
Differentially expressed outer membrane proteins (**A**), transport proteins (**B**), and iron retention proteins (**C**) in ZvL2-PA compared to ZvL2-GLU and MG1655-PA compared to MG1655-GLU. Heatmaps represent −log2 Fold Change (FC) values (*p* value ≤ 0.05). White boxes are the proteins with non–significant changes.

**Table 1 ijms-25-10118-t001:** (**A**). Membrane proteins upregulated in ZvL2 isolate grown on M9 medium supplemented with PA compared with glucose. Quantitative analysis was performed using PDQuest 8.0 software (Bio-Rad). The cutoff score for protein identification in the mascot engine was 44 (*p* < 0.05). (**B**). Membrane proteins downregulated in ZvL2 isolate grown on M9 medium supplemented with PA compared with glucose. Quantitative analysis was performed using PDQuest 8.0 software (Bio-Rad). The cutoff score for protein identification in the mascot engine was 44 (*p* < 0.05).

**(A)**			
**Protein ID**	**Protein**	**Score**	**ZvL2-PA/ZvL2-GLU**
Porins			
P0A910	Porin OmpA, isoform 1 Porin OmpA, isoform 2	158 162	10.14 ± 1.32 11.13 ± 1.45
P0A915	Porin OmpW	44	8.45 ± 1.10
P02931	Porin OmpF	110	6.15 ± 0.80
Transport			
P02943	Maltose-inducible porin LamB	77	6.43 ± 0.84
P10384	Outer membrane protein Fadl	80	3.97 ± 0.52
P02930	Outer membrane protein TolC		2.84 ± 0.37
P69797	Mannose transprter ManX	153	2.31 ± 0.30
P10346	glutamine ABC transporter ATP-binding protein GlnQ	91	3.05 ± 0.40
Biosynthesis and binding of the siderophore enterobactin			
P0ADI4	Enterochelin synthase component B EntB	179	4.45 ± 0.58
P10378	Enterochelin synthase component E EntE	108	2.66 ± 0.351
P05825	Enterobactin outer-membrane receptor FepA	141	3.26 ± 0.42
Cell division			
P0ABH0	Cell division protein FtsA	101	2.62 ± 0.34
P0A9X4	Rod shape-determining protein MreB	133	3.13 ± 0.40
Lipopolysaccharides biosynthesis			
P24174	Mannose-1-phosphate guanylyltransferase ManC	14	10.34 ± 1.24
P0A715	2-dehydro-3-deoxyphosphooctonate aldolase KdsA	79	7.18 ± 0.93
P45395	arabinose 5-phosphate isomerase KdsD	49	3.30 ± 0.43
P0AEP3	UTP-glucose-1-phosphate uridylyltransferase GalU	68	2.99 ± 0.39
ATP- synthase			
P0ABB0	ATP F0F1 synthase subunit alpha AtpA	211	1.91 ± 0.25
P0ABB4	ATP F0F1 synthase subunit beta AtpB	108	1.75 ± 0.23
Elongation factors			
P0CE47	elongation factor Tuf	194	10.14 ± 1.32
Protein kinase			
P0ACY3	serine/threonine protein kinase YeaG		3.26 ± 0.42
Diacylation of substrates			
P23908	acetylornithine deacetylase ArgE	105	1.90 ± 0.26
Catabolism of short chain fatty acids			
P36683	Aconitate hydratase B AcnB	265	12.40 ± 1.61
Reversible interconversion of acetyl-CoA and acetyl phosphate			
P0A9M8	phosphate acetyltransferase Pta	53	1.85 ± 0.24
Pyruvate metabolism			
P09373	Formate acetyltransferase PflB	158	6.34 ± 0.82
Chaperones			
P0C054	Heat Shock Protein IbpA	80	8.97 ± 1.16
P0C058	Heat Shock Protein IbpB	55	3.98 ± 0.44
Peptidoglycan recycling			
P77570	Anhydro-N-acetylmuramic acid kinase AnmK	80	1.35 ± 0.18
**(B)**			
**Gene**	**Protein**	**Score**	**ZvL2- PA/ZvL2 Glu**
Membrane protein			
P0A9101	Porin OmpA	168	0.074 ± 0.01
P0A917	Outer membrane protein X precursor OmpX	89	0.26 ± 0.03
Amino acid biosynthesis			
P2566	5-methyltetrahydropteroyltriglutamate homocysteine methyltransferase MetE	192	0.78 ± 0.10
P00904	Bifunctional Protein TrpGD	52	0.65 ± 0.08
P0A912	peptidoglycan-associated outer membrane lipoprotein Pal	46	0.34 ± 0.04
P0ABT2	DNA starvation/stationary phase protection protein Dps	56	0.42 ± 0.05

**Table 2 ijms-25-10118-t002:** (**A**). Membrane proteins upregulated in K12 MG1655 grown on M9 medium supplemented with PA compared with glucose. Quantitative analysis was performed using PDQuest 8.0 software (Bio-Rad). The cutoff score for protein identification in the mascot engine was 44 (*p* < 0.05). (**B**). Membrane proteins downregulated in K12 MG1655 grown on M9 medium supplemented with PA compared with glucose. Quantitative analysis was performed using PDQuest 8.0 software (Bio-Rad). The cutoff score for protein identification in the mascot engine was 44 (*p* < 0.05).

**(A)**			
**Gene**	**Protein**	**Score**	**MG1655- PA/MG1655-GLU**
Porins			
P0A915	Porin OmpW	45	3.99 ± 0.52
P02931	Porin OmpF	65	3.21 ± 0.38
Transport			
P02943	Maltose-inducible porin LamB	78	3.43 ± 0.44
P10384	Outer membrane protein Fadl	80	3.14 ± 0.47
P02930	Outer membrane protein TolC	55	2.82 ± 0.33
P10346	glutamine ABC transporter ATP-binding protein GlnQ	91	2.17 ± 0.17
Biosynthesis and binding of the siderophore enterobactin			
P0ADI4	Enterochelin synthase component B EntB	124	6.89 ± 0.55
P10378	Enterochelin synthase component E EntE	98	2.18 ± 0.28
P05825	Enterobactin outer-membrane receptor FepA	208	2.53 ± 0.30
Cell division			
P0A9X4	Rod shape-determining protein MreB	44	5.57 ± 0.50
Lipopolysaccharides biosynthesis			
P0AEP3	UTP–glucose-1-phosphate uridylyltransferase GalU	68	4.54 ± 0.44
ATP-synthase			
P0ABB0	ATP F0F1 synthase subunit alpha AtpA	211	1.99 ± 0.23
P0ABB4	ATP F0F1 synthase subunit beta AtpB	190	1.88 ± 0.22
Elongation factors			
P0CE47	elongation factor Tuf	194	8.44 ± 0.68
Protein kinase			
P0ACY3	serine/threonine protein kinase YeaG	65	3.17 ± 0.42
Catabolism of short chain fatty acids			
P36683	Aconitate hydratase B AcnB	265	5.23 ± 0.49
Pyruvate metabolism			
P09373	Formate acetyltransferase PflB	174	6.33 ± 0.56
Chaperones			
P0C054	Heat Shock Protein IbpA	80	4.19 ± 0.50
P0C058	Heat Shock Protein IbpB	44	1.99 ± 0.20
**(B)**			
**Gene**	**Protein**	**Score**	**MG1655-PA/MG1655-GLU**
Membrane proteins			
P0A910	Porin OmpA, isoform 1 Porin OmpA, isoform 6	130 129	0.57 ± 0.07 0.03 ± 0.01
P34210	Outer membrane protein X precursor OmpX	89	0.43 ± 0.06
Amino acid biosynthesis			
P25665	5-methyltetrahydropteroyltriglutamate homocysteine methyltransferase MetE	192	0.58 ± 0.10
P0ADE6	K(+) binding protein Kbp	149	0.17 ± 0.02

## Data Availability

The obtained proteomic data were upload in PRIDE (project accession: PXD054715, project doi:10.6019/PXD054715).

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
