# Peer review of "The Role of Propionate-Induced Rearrangement of Membrane Proteins in the Formation of the Virulent Phenotype of Crohn’s Disease-Associated Adherent-Invasive Escherichia coli"

_ijms, 2024, doi:10.3390/ijms251810118_

Round 1

Reviewer 1 Report

Comments and Suggestions for Authors

The scientific approach and conclusions of the manuscript are valid, even if the methodology is a bit limited. The manuscript suffers a bit from being very verbose. While the introduction is very good, informative, and easy to parse, some parts of the results and discussion sections are very convoluted. It is sometimes difficult to understand the point that the authors want to make. The results would thus benefit much from a more structured approach, also with some key conclusions inserted after one of the major discussion points. Indeed, there is little connection between the results chapters and they would benefit from some connecting sentences to explain the rationale of the authors.

In addition to these general comments, I have the following specific comments

·         The abstract would benefit from slight structuring into something like “Background”, “Results”, “Conclusions”

·         Many of the paragraphs in the results section, especially 2.3.1 (see also comment on numbering below), is presented as one monolithic blocm of text which makes it hard to read. I suggest to structure these sections more, e.g. with paragraph breaks and subheadings that actually identify the relevance of the findings.

·         The same is true for the discussion section. This can be shortened and should be structured using subheadings.

·         In addition, I suggest to re-name the results sections’ headings to something more descriptive that directs the reader to the main point the authors want to make, instead of just calling it after the respective method (“2D-DIGE”, “OmpA”, etc.).

·         Parts of the results section read more like the discussion section rather than the results. I suggest either combining results and discussion into one “Results and Discussion” or moving some of the interpretation from the current results (Which will also help with the structure of the results, see previous comments) to the discussion section and focusing in the results section only on the description of the results in a systematic way.

·         Figure 4 is currently only mentioned briefly in the text. I suggest to add a paragraph expounding on the results of this figure. The figure legend also misses an explanation what is shown here (are these mean fold changes +/- SD)? Histograms in general are not considered state-of-the art anymore and should be replaced with individual values (or least show the individual values as overlay in addition to the bars).

·         The same applies to figure 1: please show individual data points instead of bar charts (compare: doi:10.1371/journal.pbio.1002128)

·         Discussion section: It would be valuable if the authors would also indicate the limitations of their study in a spate small paragraph at the end of this section.

·         Line 629: I am not sure that this conclusion can be drawn if no co-culture experiments have been made. Adding at least a small experiment on would greatly enhance the significance of the paper.

·         Why is the conclusion twice in the manuscript? Once starting with line 628, and another time at lines 777. The statements are also not the same. Please revise.

·         I do not understand the data processing section in the methods. The authors say: “Statistical significance of differences between groups was assessed using Student's t-test for independent variables and Mann–Whitney test.” There are a few unclear things here: 1) It is not clear from the figure legends or the text which statistical method was unsed on which dataset. A t-test requires normal disctributed data, a Mann-Whitney U test is for non-normal distributed data. It is not indicated whether the data were tested for normal distribution, i.e. how the respective test was selected. 2) However, it is actually unclear which data the authors have analyzed with a t-test or Mann Whitney U test anyway. Both tests compare two groups. The only place where these tests are mentioned in the figure legend are Figure 1. However, in figure 1, three groups are compared – the correct test to compare three or more groups is an ANOVA (for normal distributed data) or a Kruskall-Wallis Test (for non-normal distributed data).

·         This leads to another problem with figure 1: if for some results Mann-Whitney U was used, this indicates that the data were not normally distributed. If the data were not normal distributed, the data should not be presented as mean+-SD but as median +- IQR.

·         While it is stated that experiments were performed “in accordance with the requirements of the local Ethic Committee”, please provide the number of the Ethics Committee permit for this study as well as the actual name / residence of the Ethics Committee.

Minor comments

·         The first subsection of 2.3 is incorrectly named “2.3.12 D-DIGE”. This should probably be 2.3.1 2D-DIGE”.

·         Same for 4.5.2

·         Parts of section 2.3.1 are incorrectly written in the present tense (results should always be written in the past tense), ca. lines 185-196. Please revise.

·         Currently, both “Fig.” and “Figure” are used in the text. Please revise for consistency.

·         Why is the paragraph from line 309 onwards printed in bold face?

·         Figure 8: please indicate units for the heatmap legend.

·         E.g. line 487: suggest to change “cells” here with “bacteria” otherwise this is a bit confusing.

·         Line 623: “and capsules” or “i.e. capsules”?

·         Why is section 4.11 in italics?

·         Line 771: Why is there an “?” after “Origin?”

·         Why are there colored highlights on the front page?

Comments on the Quality of English Language

The manuscript is generally well written, with only minor language editing necessary. I made some specific comments above. The manuscript is very wordy though and both results and dcisussion section could easily be cut by around 25%. At the very least, both sections should be structured with more subheadings.

Author Response

Thank you very much for your attention to our manuscript. All corrections are highlighted in color.

Comment 1     The abstract would benefit from slight structuring into something like “Background”, “Results”, “Conclusions”

Response 1     We structured the abstract as you suggested, lines 16-34.

Comment 2     Many of the paragraphs in the results section, especially 2.3.1 (see also comment on numbering below), is presented as one monolithic blocm of text which makes it hard to read. I suggest to structure these sections more, e.g. with paragraph breaks and subheadings that actually identify the relevance of the findings. The same is true for the discussion section. This can be shortened and should be structured using subheadings.

Response 2     We have tried to structure the sections in more detail using paragraphs and subheadings.

Comment 3     In addition, I suggest to re-name the results sections’ headings to something more descriptive that directs the reader to the main point the authors want to make, instead of just calling it after the respective method (“2D-DIGE”, “OmpA”, etc.).

Response 3     We have renamed the results sections’ headings (lines 176-178, 290-291, 339-342, 502, 514)       

Comment 4        Parts of the results section read more like the discussion section rather than the results. I suggest either combining results and discussion into one “Results and Discussion” or moving some of the interpretation from the current results (Which will also help with the structure of the results, see previous comments) to the discussion section and focusing in the results section only on the description of the results in a systematic way.

Response 4     We have combined the results and discussion sections into one section.

Comment 5        Figure 4 is currently only mentioned briefly in the text. I suggest to add a paragraph expounding on the results of this figure. The figure legend also misses an explanation what is shown here (are these mean fold changes +/- SD)? Histograms in general are not considered state-of-the art anymore and should be replaced with individual values (or least show the individual values as overlay in addition to the bars).

Response 5     For clarity, we have replaced Fig. 4 with Tables 1 and 2, since the data presented in Fig. 4 are nothing more than comparative quantitative analysis of protein spots corresponding to differentially expressed proteins in two-dimensional maps. We believe that individual values ​​will be more understandable and visual (lines 273-289).

Comment 6        The same applies to figure 1: please show individual data points instead of bar charts (compare: doi:10.1371/journal.pbio.1002128)

Response 6 We have changed Figure 1 to box-and-whiskers diagrams; individual data points can be seen in Figure S1.

Comment 7       Discussion section: It would be valuable if the authors would also indicate the limitations of their study in a spate small paragraph at the end of this section.

Response 7     We have indicated the limitations of our study at the end of the results and discussion section, lines 661-663.

Comment 8       Line 629: I am not sure that this conclusion can be drawn if no co-culture experiments have been made. Adding at least a small experiment on would greatly enhance the significance of the paper.

Response 8     We did not need to conduct co-cultivation experiments. Your question is not very clear. Perhaps the text in the conclusion was not written very clearly. We have replaced part of the text in the conclusion to make it clearer. Lines 649-658.

Comment 9        Why is the conclusion twice in the manuscript? Once starting with line 628, and another time at lines 777. The statements are also not the same. Please revise.

Response 9     We have revised it, the conclusion is now in one place, lines 649-658.

Comment 10         I do not understand the data processing section in the methods. The authors say: “Statistical significance of differences between groups was assessed using Student's t-test for independent variables and Mann–Whitney test.” There are a few unclear things here: 1) It is not clear from the figure legends or the text which statistical method was unsed on which dataset. A t-test requires normal disctributed data, a Mann-Whitney U test is for non-normal distributed data. It is not indicated whether the data were tested for normal distribution, i.e. how the respective test was selected. 2) However, it is actually unclear which data the authors have analyzed with a t-test or Mann Whitney U test anyway. Both tests compare two groups. The only place where these tests are mentioned in the figure legend are Figure 1. However, in figure 1, three groups are compared – the correct test to compare three or more groups is an ANOVA (for normal distributed data) or a Kruskall-Wallis Test (for non-normal distributed data).

Response 10   Finally, we used Mann-Whitney test comparing bacteria-GLU with those grown with PA (difference is marked with *). In some diagrams we show also difference from control for bacteria-PA (#). Moreover, we have 4 values for each experimental point so it is difficult to apply Kruskall-Wallis test here.

Comment 11         This leads to another problem with figure 1: if for some results Mann-Whitney U was used, this indicates that the data were not normally distributed. If the data were not normal distributed, the data should not be presented as mean+-SD but as median +- IQR.

Response 11   We have changed figure 1 to box-and-whiskers diagrams.

Comment 12         While it is stated that experiments were performed “in accordance with the requirements of the local Ethic Committee”, please provide the number of the Ethics Committee permit for this study as well as the actual name / residence of the Ethics Committee.

Response 12   We have provided the ethics committee approval number for this study, as well as the actual name/location of the ethics committee, lines 675-682.

Minor comments

Comment 1         The first subsection of 2.3 is incorrectly named “2.3.12 D-DIGE”. This should probably be 2.3.1 2D-DIGE”.

Response 1     We have made corrections, line 176.

Comment 2         Same for 4.5.2

Response 2     We have made corrections, line 712.

Comment 3        Parts of section 2.3.1 are incorrectly written in the present tense (results should always be written in the past tense), lines 185-196. Please revise.

Response 3     We have made corrections throughout the text.

Comment 4         Currently, both “Fig.” and “Figure” are used in the text. Please revise for consistency.

Response 4     We have made corrections.

Comment 5         Why is the paragraph from line 309 onwards printed in bold face?

Response 5     We have made corrections, line 322.

Comment 6         Figure 8: please indicate units for the heatmap legend.

Response 6     We have made corrections, line 512.

Comment 7         E.g. line 487: suggest to change “cells” here with “bacteria” otherwise this is a bit confusing.

Response 7     We have made corrections, line 501.

Comment 8         Line 623: “and capsules” or “i.e. capsules”?

Response 8     Thus, the difference between virulent ZvL2-PA and non-virulent ZvL2-GLU may be related to the activation of Rcs and PhoP/Q signals, which promote the formation of defense mechanisms, leading to the formation of biofilms as well as capsules, line 644.

Comment 9         Why is section 4.11 in italics?

Response 9     We have made corrections, line 802.

Comment 10         Line 771: Why is there an “?” after “Origin?”

Response 10   We have made corrections.

Comment 11         Why are there colored highlights on the front page?

Response 11   We have made corrections. These are editorial changes, apparently.

Reviewer 2 Report

Comments and Suggestions for Authors

The original article by Pobeguts et al. entitled “The role of propionate-induced rearrangement of membrane proteins in the formation of the virulent phenotype of Crohn's Disease-associated adherent-invasive Escherichia coli.” investigates whether ZvL2 AIEC through proprionate (PA) activation is associated with membrane proteins modification compared to ZvL2 grown on glucose and K12 MG1655. 2D gel electrophoresis and liquid chromatography–mass spectrometry were used. Differentially expressed proteins of zvL2 and K12 MG1655 were identified and categorized. amino acid sequences of OmpA from both strains were analyzed.  

Introduction

  • CD is NOT defined as a granulomatous disease; please modify
  • AIEC can also interact with dendritic cells; please modify
  • Regarding genes associated with the AIEC pathotype, I would suggest the authors have a look at this manuscript 10.1080/19490976.2024.2380064
  • Please include this abbreviation Liquid chromatography–mass spectrometry

Results

  • I would suggest the authors reduce figures (4-6 in one figure?; figure 5 as supplementary?)
  • What do Figure 1 experiments add to your previous studies?

Discussion

  • Limitations of the study?
  • The authors focus only on ZvL2, limiting the generalizability of the findings. A second AIEC strain would be ideal to confirm whether the observed effects are consistent across different isolates of AIEC associated with CD. If not, this should be addressed as a study limitation.
  • besides changes associated with PA, no mechanistic insights are provided into how these changes translate into increased virulence at the molecular level. 
  •  Do in vitro conditions replicate gut human ones in CD patients or murine models?
Comments on the Quality of English Language

can be improved

Author Response

Thank you very much for your attention to our manuscript. All corrections are highlighted in color.

Introduction

Comment 1:      CD is NOT defined as a granulomatous disease; please modify.

Response 1        We have made corrections, line 42.

Comment 2       AIEC can also interact with dendritic cells; please modify

Response 2        Yes, indeed, we have made corrections, line 44.

Comment 3       Regarding genes associated with the AIEC pathotype, I would suggest the authors have a look at this manuscript   10.1080/19490976.2024.2380064

Response 3        Thank you very much for this manuscript, very interesting work, I have not seen it yet. We have referred to it in our manuscript, lines 44, 603-609.

Comment 4       Please include this abbreviation Liquid chromatography–mass spectrometry

Response 4        We have made corrections, line 102.

Results

Comment 1           I would suggest the authors reduce figures (4-6 in one figure?; figure 5 as supplementary?)

Response 1            We have combined the results and discussion sections into one section.

Comment 2           What do Figure 1 experiments add to your previous studies?

Response 2           As in the previous work, we show that the abilities of AIEC cultured on propionate differ from the abilities of AIEC cultured on glucose. Moreover, propionate activates, and glucose, on the contrary, reduces the virulence properties of AIEC. Previously, we showed that propionate activates adhesion-invasion and survival in macrophages, and now we show that propionate activates neutrophils, unlike glucose, which reduces these properties.

Discussion

Comment 1           Limitations of the study? The authors focus only on ZvL2, limiting the generalizability of the findings. A second AIEC strain would be ideal to confirm whether the observed effects are consistent across different isolates of AIEC associated with CD. If not, this should be addressed as a study limitation.

Response 1          We agree with you completely. We have indicated the limitations of our study at the end of the results and discussion section, lines 661-663.

Comment 2           Besides changes associated with PA, no mechanistic insights are provided into how these changes translate into increased virulence at the molecular level. 

Response 2          Well, why? We show conclusively that propionate causes a rearrangement of the abundance of membrane proteins, which leads to the activation of Rcs and PhoP/Q signaling pathways and changes in the composition and/or modification of LPS, which may be directly related to neutrophil activation. In addition, we also show that the porin OmpA may be involved in neutrophil activation, OmpA is a major porin and a multifunctional protein. These changes may also play a role in the adhesion-invasion and survival capabilities of macrophages. We have previously shown that propionate causes a significant increase in these capabilities in AIEС. We hypothesize mechanisms for the activation of AIEC virulence properties by propionate, further experiments are needed for this, and we are currently working on this.

Comment 3          Do in vitro conditions replicate gut human ones in CD patients or murine models?

Response 3             No, of course they don't. But the concentration of propionate we use is consistent with the concentration in the human intestine, which ranges from 1.7 mM in the ileum to 27 mM in the colon (Cummings et al., 1987, Hung et al., 2013).

Round 2

Reviewer 2 Report

Comments and Suggestions for Authors

All comments clear